# Assessment of the Efficacy of the Antihistamine Drug Rupatadine Used Alone or in Combination against Mycobacteria

**DOI:** 10.3390/pharmaceutics16081049

**Published:** 2024-08-07

**Authors:** Xirong Tian, Wanli Ma, Buhari Yusuf, Biyi Su, Jinxing Hu, Tianyu Zhang

**Affiliations:** 1State Key Laboratory of Respiratory Disease, Guangdong-Hong Kong-Macao Joint Laboratory of Respiratory Infectious Diseases, China-New Zealand Joint Laboratory on Biomedicine and Health, Guangzhou Institutes of Biomedicine and Health (GIBH), Chinese Academy of Sciences (CAS), Guangzhou 510530, China; tian_xirong@gibh.ac.cn (X.T.); ma_wanli@gibh.ac.cn (W.M.); yusuf@gibh.ac.cn (B.Y.); 2University of Chinese Academy of Sciences (UCAS), Beijing 100049, China; 3State Key Laboratory of Respiratory Disease, Guangzhou Chest Hospital, Guangzhou 510095, China; muyezi-888@163.com

**Keywords:** mycobacteria, drug resistance, rupatadine, drug repurposing, nonreplicating

## Abstract

The emergence of drug-resistant mycobacteria has rendered many clinical drugs and regimens ineffective, imposing significant economic and healthcare burden on individuals and society. Repurposing drugs intended for treating other diseases is a time-saving, cost-effective, and efficient approach for identifying excellent antimycobacterial candidates or lead compounds. This study is the first to demonstrate that rupatadine (RTD), a drug used to treat allergic rhinitis, possesses excellent activity against mycobacteria without detectable resistance, particularly *Mycobacterium tuberculosis* and *Mycobacterium marinum*, with a minimal inhibitory concentration as low as 3.13 µg/mL. Furthermore, RTD exhibited moderate activity against nonreplicating *M. tuberculosis* with minimal inhibitory concentrations lower than drugs targeting the cell wall, suggesting that RTD has great potential to be modified and used for the treatment of nonreplicating *M. tuberculosis*. Additionally, RTD exhibits partial synergistic effects when combined with clofazimine, pretomanid, and TB47 against *M. tuberculosis*, providing the theoretical foundation for the development of treatment regimens. Transcriptomic profiling leads us to speculate that eight essential genes may be the targets of RTD or may be closely associated with mycobacterial resistance to RTD. In summary, RTD may be a promising hit for further antimycobacterial drug or regimen optimization, especially in the case of nonreplicating mycobacteria.

## 1. Introduction

*Mycobacterium tuberculosis*, a highly pathogenic bacterium within the mycobacterium genus, is the causative agent of tuberculosis (TB), posing a serious threat to global health [1]. The challenge posed by *M. tuberculosis* remains formidable, with an estimated 10 million new TB cases and 1.5 million TB deaths annually [2]. Misuse of antibiotics and poor patient compliance contribute significantly to the difficulty of completely eradicating this pathogenic bacterium [3]. These challenges lead to the emergence of drug-resistant strains, further complicating this issue and rendering traditional therapeutic regimens ineffective [4]. Consequently, the most crucial objective in the current treatment of drug-resistant TB is to search for more effective anti-*M. tuberculosis* drugs and regimens.

In addition to *M. tuberculosis* and *Mycobacterium leprae*, there are approximately 150 species of opportunistic pathogenic nontuberculous mycobacteria (NTM) that can cause lung diseases and skin infections in adults, as well as cervical lymphadenitis in children [5]. NTM possess a distinctive cell wall structure characterized by a thin layer of peptidoglycan surrounded by a thick lipid-rich outer layer [6]. This unique composition of the cell wall enables NTM to adhere to rough surfaces, resist antibiotics and disinfectants, and survive in low-oxygen and other stressful conditions [5,7]. Many NTM, such as *Mycobacterium abscessus*, demonstrate intrinsic resistance to numerous clinical drugs, complicating and prolonging the treatment of NTM infections [8]. Therefore, there is an urgent need for the discovery and development of effective drugs and regimens for clinical anti-NTM therapy. One practical approach is repurposing existing drugs that were originally designed for treating other diseases. These drugs have already been used clinically, which can significantly reduce the time required for them to be developed into the commercialized antimycobacterial drugs [9]. Moreover, these drugs can also be further modified into new drugs against mycobacteria as lead compounds.

Allergic rhinitis is a commonly prevalent disease, affecting 10–25% of the global population and significantly impacting patients’ daily lives [10]. Histamine plays a crucial role in the development of allergic rhinitis, primarily through its interaction with the histamine H1 receptor [11]. Additionally, the platelet-activating factor is another significant inflammatory factor that promotes the release of histamine and vice versa in various tissues and cells [11]. Rupatadine (RTD) is a second-generation H1-antihistamine agent with dual affinity for both histamine H1 and platelet-activating factor receptors, unlike most clinical drugs that only inhibit a single inflammatory factor [12]. RTD is recommended for patients aged 12 years and older who suffer from seasonal allergic rhinitis, perennial allergic rhinitis, and chronic idiopathic urticaria due to its fast-acting and long-lasting effect [13,14]. Unlike first-generation H1 antihistamines, RTD does not cause side effects such as drowsiness, fatigue, headache, memory and learning difficulties, and visual disturbances [15].

From the discovery of RTD to date, there have been no reports on the activity of RTD against mycobacteria. In this study, we discovered unexpected activity of RTD against mycobacteria, suggesting that RTD has great potential as an antimycobacterial drug candidate or as a lead compound for optimization.

## 2. Materials and Methods

### 2.1. Bacteria and Culture Conditions

The mycobacteria used in this study were preserved at −80 °C and cultured at 37 °C in 7H9 broth supplemented with 10% oleic acid-albumin-dextrose-catalase enrichment medium, 0.2% glycerol, and 0.05% Tween 80. The autoluminescent mycobacterial strains, including autoluminescent *M. tuberculosis* H37Ra (AlRa), *M. tuberculosis* H37Rv (AlRv), *Mycobacterium marinum* (AlMm), *M. abscessus* (AlMab), and *Mycobacterium smegmatis* (AlMs), were engineered and preserved by our laboratory [16,17,18]. It has been verified that the insertion of the *lux* gene cluster had no effect on drug susceptibility or growth rate of autoluminescent mycobacteria compared to their wild-type counterparts [16,17,18]. Autoluminescent mycobacteria possess the distinctive ability to emit blue–green light without the supplement of additional substrate. This characteristic enables relative light units (RLUs) to serve as a surrogate for colony-forming units when evaluating drug activity against mycobacteria.

### 2.2. Antimicrobials

RTD was bought from AiYan (Shanghai, China) with a specified purity of 98%. Amikacin (AMK), clofazimine (CLO), clarithromycin (CLR), isoniazid (INH), levofloxacin (LEV), linezolid (LZD), pretomanid (PTM), rifampicin (RIF), and streptomycin (STR) were bought from Meilun (Dalian, China). TB47, an imidazopyridine amide antibiotic targeting the QcrB subunit within the cytochrome *bc1* oxidase complex of electron transfer chain, was synthesized by Boji (Guangzhou, China) [19]. INH, AMK, and STR were dissolved in sterile water, whereas the remaining drugs were dissolved in dimethyl sulfoxide (Xilong, Shantou, China). All drugs were diluted by their corresponding solvents and stored at −20 °C until use.

### 2.3. Evaluating Activity of RTD against Actively Growing Mycobacteria

The strains AlRa, AlRv, AlMm, AlMab, and AlMs were cultured until their optical density at 600 nm (OD_600_) reached 0.8, with the luminescence peaking at approximately 5 × 10^6^ RLUs/mL. The cultures were then diluted to 10^4^ RLUs/mL using 7H9 without Tween 80. Following dilution, 196 µL of the bacterial culture and 4 µL of the drug solution were thoroughly mixed in the same sterile tube. The RLUs of the mixture were monitored at regular intervals over the following hours or days. Each experiment was conducted in triplicate. The minimal inhibitory concentration (MIC) based on the autoluminescence values is defined as the lowest drug concentration that reduces the RLUs to less than or equal to 10% of the RLUs detected in the solvent-treated control group [20].

To determine the MIC of RTD against non-luminescent *M. tuberculosis* H37Rv, a microdilution method was employed in combination with a microplate Alamar Blue assay (MABA) [21]. Following a 7-day co-incubation, a mixture consisting of 20 µL of Alamar Blue and 12.5 µL of 20% Tween 80 was added into each well of a 96-well plate. The plate was then incubated at 37 °C in a thermostat, and the color change was monitored after a 24 h period. A transition from blue–purple to pink of Alamar Blue indicates bacterial growth. Therefore, the MIC is identified as the lowest drug concentration corresponding to the blue–purple wells [21].

### 2.4. Evaluating the Activity of RTD against Nonreplicating M. tuberculosis in Diverse Media

Following the protocols of the recently developed low-oxygen-recovery model, AlRa was cultured under aerobic conditions until the OD_600_ reached 0.6–0.8 and RLUs increased to approximate 5 × 10^6^/mL [22]. Methylene blue was added into AlRa culture at a finial concentration of 6 μg/mL. The AlRa bacteria were then cultured in a low-oxygen incubator until the blue color of methylene blue indicator disappeared, indicating that the bacterial culture had achieved a nonreplicating state under anaerobic conditions. A mixture of 196 µL of 10-fold diluted nonreplicating AlRa (using 7H9 without Tween 80 and supplemented with glycerol or cholesterol) and 4 μL of RTD was added to the same sterile tube. After seven days of incubation, activated carbon (in a volume ratio of 1:5, 50 μL into 200 μL) was added to eliminate any potential carryover effects of residual drug [19,23]. The tubes were transferred to aerobic conditions for recovery, and their RLUs were measured at 7 h intervals over the subsequent 28 h. The experiment was conducted in triplicate.

### 2.5. Evaluating the Activities of RTD in Combination with Anti-TB Drugs

To assess the efficacy of the combination of RTD with other anti-TB drugs against *M. tuberculosis*, a checkerboard assay was conducted. The drugs used in combination with RTD included LEV, AMK, INH, STR, PTM, RIF, CLO, LZD, and TB47. Each drug was evaluated at six different concentrations derived from their previously determined MICs. Specifically, 2 µL of RTD, 2 µL of the chosen drug solution, and 196 µL of a diluted bacterial culture at 1.5 × 10^5^ RLUs/mL were added to individual wells of a 96-well plate. Subsequently, the plates were incubated at 37 °C for seven days. After incubation, the RLUs were measured using a luminometer. The fractional inhibitory concentration index (FICI) was determined by calculating the ratio of the MIC of RTD in combination with the MIC of RTD alone, plus the MIC of selected drugs in combination with the MIC of selected drugs alone [24,25]. The effects were classified into five groups based on the FICI: synergistic (FICI ≤ 0.5), partially synergistic (0.5 < FICI < 1), additive (FICI = 1), irrelevant (1 < FICI ≤ 4), and antagonistic (FICI > 4) [24].

### 2.6. Evaluating the In Vivo Anti-M. tuberculosis Activity of RTD

Female BALB/c mice, aged 6–8 weeks, underwent a five- to seven-day acclimatization period prior to the initiation of experimental procedures. Subsequently, all mice were exposed to 10 mL of AlRv culture (10^6^ RLUs/200 μL) using an inhalation exposure system.

The in vivo activity of RTD against *M. tuberculosis* was initially assessed using oral administration. The dosages of RTD were set as 6.25 and 25 mg/kg. The combination of RTD and TB47 demonstrated a partial synergistic effect against *M. tuberculosis* in vitro, with the FICI being 0.75 as assessed previously. Consequently, the in vivo efficacy of the combination of RTD with TB47 (25 mg/kg) was also evaluated. Mice were treated with drugs or solvent on the eighth day post-infection. The treatments were administered daily for eight days until the live mice RLUs of the solvent group exceeded 400. All mice were sacrificed for the measurement of lung suspension RLUs at the initiation and one day after completion of treatment.

Meanwhile, we also employed a recently developed noninvasive murine model of inhalable administration to assess the in vivo anti-*M. tuberculosis* activity of RTD [26]. RTD is a less soluble compound that can reach a maximum concentration of 1.56 mg/mL (3.75 mM). Briefly, the infected mice were treated daily with either 4 mL of RTD at a concentration of 1.56 mg/mL (3.75 mM) or distilled water via inhalation [26]. Meanwhile, the RIF was used as the positive control, with a concentration of 2 mg/mL (4.81 mM). The duration of inhalable administration (4 mL) each time was approximately 25 min. The treatment duration lasted for 15 days. The in vivo anti-*M. tuberculosis* activity of RTD was determined by comparing the live mice RLUs and lung suspension RLUs of both RTD-treated and untreated groups.

### 2.7. The Transcriptomic Profiling of RTD-Treated and Untreated M. tuberculosis

The AlRa was transferred into fresh 7H9 broth at a volume ratio of 1:10. Based on the MIC of RTD against *M. tuberculosis*, RTD was added to achieve a sub-inhibitory concentration concurrently. The bacteria were cultured with and without RTD and incubated in a shaker at 37 °C until the OD_600_ reached approximately 0.6–0.8. All samples were sent to Jingnuo (Shanghai, China) for transcriptome sequencing.

### 2.8. Statistical Analysis

Log-transformed data were analyzed for drug efficacy using GraphPad Prism version 8.3.0. A two-way analysis of variance was performed, with statistical significance set at *p* < 0.05.

## 3. Results

### 3.1. Antimycobacterial Activity of RTD

We initially bought a library of FDA-approved compounds, comprising approximately 2000 clinical and preclinical drugs. We then utilized an autoluminescent *M. tuberculosis* H37Ra strain to primarily screen drugs with potential anti-TB activity, employing only three concentrations (100, 10, and 1 μg/mL). Notably, we found that RTD exhibited significant antimycobacterial activities and further examined its efficacy when used alone or in combination against mycobacteria.

This study is the first to investigate the efficacy of RTD against mycobacteria, pioneering its exploration in this field. As depicted in Figure 1 and Table 1, RTD exhibited substantial bacteriostatic activity against AlRa, AlMm, AlMab, and AlMs at concentrations of 3.13, 3.13, 50, and 12.5 µg/mL, respectively. The MICs of RTD against AlMab and AlMs were relatively higher compared to those against AlRa and AlMm. RTD potently inhibited the growth of AlRv, a virulent laboratory strain, at a remarkably low concentration of 3.13 μg/mL (Figure 1). However, the MIC of RTD against non-luminescent *M. tuberculosis* H37Rv was 25 μg/mL when assessed by MABA. Meanwhile, RTD exhibited no activity against certain isolates of *M. tuberculosis*, as evidenced by MICs surpassing 50 μg/mL. Additionally, we also evaluated the activities of RTD against prevalent Gram-positive and Gram-negative bacteria, including *Klebsiella pneumoniae*, *Pseudomonas aeruginosa*, *Enterococcus faecalis*, and *Staphylococcus aureus*. Nevertheless, the MICs of RTD against all the mentioned Gram-positive and Gram-negative bacteria exceeded 100 μg/mL.

### 3.2. Activity of RTD against Nonreplicating AlRa Under Diverse Conditions

Using the recently established low-oxygen-recovery model, we assessed the efficacy of RTD against AlRa under four distinct conditions: aerobic and anaerobic conditions using 7H9 medium supplemented with glycerol or aerobic and anaerobic conditions using 7H9 medium supplemented with cholesterol [22]. Figure 2 and Table 2 show the activity of RTD against *M. tuberculosis* under diverse conditions. The MICs of RTD were consistent under the aerobic conditions, whether the medium was enriched with glycerol or cholesterol, with respective MICs of 3.13 µg/mL and 1.56 µg/mL (Figure 2A). However, under anaerobic conditions, the MICs were 25 µg/mL and 50 µg/mL when the liquid medium was supplemented with either glycerol or cholesterol, respectively (Figure 2B). As expected, the MICs of RTD against nonreplicating AlRa were found to be higher than those against actively growing AlRa.

### 3.3. Partial Synergistic Effect of RTD in Combination with Several Antimycobacterial Drugs against M. tuberculosis

We investigated the potential synergistic activities of RTD in combination with various antimycobacterial drugs against *M. tuberculosis*, including AMK, CLO, INH, LEV, LZD, PTM, RIF, STR, and TB47. As shown in Table 3, RTD exhibited partial synergistic effects with PTM, CLO, and TB47, as evidenced by FICIs of 0.5625, 0.75, and 0.75, respectively (Table 3). Notably, in the presence of sub-inhibitory concentrations of CLO, TB47, or PTM, the effective concentrations of RTD decreased to 1/4, 1/4, and 1/16 of its MIC, respectively (Table 3). The combinations of RTD with the remaining six drugs exhibited either additive or indifferent effect against *M. tuberculosis*, with all FICIs being equal to or greater than 1 (Table 3). However, no antagonistic interactions were observed between RTD and the tested antimycobacterial drugs.

### 3.4. The In Vivo Anti-M. tuberculosis Activity of RTD

In vivo evaluation of RTD was conducted using the oral administration method first, as RTD is an oral drug, with the dosages of 6.25 and 25 mg/kg. However, the results indicated that RTD did not exhibit in vivo activity against *M. tuberculosis* even at the highest dosage of 25 mg/kg (Figure 3A). The live mice RLUs of the combination of RTD with TB47 were slightly lower than those of the TB47-alone group (Figure 3A, *p* > 0.05). However, the RLUs of lung suspension showed no significant difference between the combination of RTD with TB47 and the TB47-alone groups (Figure 3A, *p* > 0.05).

To improve the lung local concentration of RTD, a noninvasive inhalation murine model was employed. The results demonstrated that the live mice RLUs and lung suspension RLUs of the RTD-treated group were not lower but significantly higher than those of the solvent-treated group (Figure 3B, *p* < 0.05 and *p* < 0.01, respectively). The RLUs of live mice and lung suspension of the RIF-treated group (2 mg/mL) were significantly lower than those of the solvent group (Figure 3B, both *p* < 0.0001). All results suggest that RTD at a concentration of 1.56 mg/mL (3.75 mM) did not exhibit significant in vivo anti-*M. tuberculosis* activity via inhalation delivery.

### 3.5. Transcriptome Profile of RTD-Treated and Untreated M. tuberculosis

We conducted multiple trials (exceeding five biological replicates for each strain) to isolate spontaneous RTD-resistant mutants using both *M. tuberculosis* and *M. marinum*. Despite incorporating 5-Bromouracil into bacterial cultures to increase the spontaneous mutation rates and plating on agar containing RTD at concentrations equivalent to only four times the MIC, our attempts were unsuccessful [27]. Hence, we could not utilize whole genome sequencing of laboratory-generated drug-resistant mutants to determine the mechanism of action of RTD. We performed a transcriptome analysis of *M. tuberculosis* in the presence and absence of RTD treatment to identify genes potentially associated with its mechanism of action. As shown in Figure 4A, out of the total genes analyzed, 1806 genes exhibited no significant changes, while the expressions of 1134 genes were upregulated and 1195 genes were downregulated. Figure 4B highlights the top three categories with the highest number of genes, namely carbon metabolism pathways, ABC transporters, and the two-component systems.

All genes with a |log_2_Foldchange|greater than 1 are listed in the Appendix A. Among these genes, *Rv0251c*, a nonessential gene, exhibited the most significant change in expression. Rv0251 is believed to be involved in the initiation step of translation at high temperature [28]. In the groups treated with RTD, the expression level of *Rv0251c* was nearly 30 times higher compared to the groups without RTD treatment. Additionally, following RTD treatment, the expression of *Rv1405c* and *Rv3161c* was upregulated, with |log_2_FoldChange| values of 4.15 and 4.05, respectively. *Rv1405c* is known to participate in adaptive processes, while *Rv3161c* primarily encodes a putative oxygenase [29,30]. Notably, *Rv0251c*, *Rv1405c*, and *Rv3161c* are all nonessential genes. Given the low probability of these genes being the targets of RTD, we expanded our analysis to include additional genes by lowering the threshold of |log_2_Foldchange| and emphasizing their essentiality as the criterion. Eight genes were filtered out based on the basic standards. They are *Rv0350*, *Rv0351*, *Rv0352*, *Rv0384c*, *Rv0440*, *Rv2720*, *Rv2827c*, and *Rv3260c* (Table 4). Among them, only the expression of *Rv3260c* gene was downregulated and the expression of the remaining seven genes were upregulated (Table 4). The gene cluster, encompassing *Rv0350, Rv0351*, and *Rv0352*, showed an over 8-fold increase in expression level following treatment for *M. tuberculosis* with RTD.

## 4. Discussion

Mycobacteria, recognized as some of the most virulent pathogens, are responsible for a wide range of human and animal diseases, including TB, leprosy, Buruli ulcer, and NTM infections [31]. The emergence of drug-resistant mycobacteria urgently needs the development of novel and potent drugs or treatment regimens, which is imperative for both individuals and society [32]. Repurposing existing drugs, originally intended for other diseases, presents a time-efficient, cost-effective, and labor-saving approach to overcome challenges associated with the screening and development of novel antimycobacterial agents.

RTD, a compound traditionally employed in the treatment of allergic rhinitis, has been identified in our study as a novel agent with potential for the development of antimycobacterial drugs. This is the first report to discover and document the efficacy of RTD against mycobacteria, including AlRa, AlRv, and AlMm, exhibiting MICs as low as 3.13 μg/mL (Figure 1). However, the MIC of RTD against wild-type *M. tuberculosis* H37Rv is 25 μg/mL, determined by MABA. As described in the reference, the emission of blue–green light by AlRv is dependent on the presence of oxygen [16]. Hence, the observed differences in the efficacy of RTD against luminescent and non-luminescent *M. tuberculosis* H37Rv may be attributed to its impact on the respiration of bacteria or other substrates needed for the reaction catalyzed by LuxAB. These results suggest that the method based on RLU determination is more sensitive than MABA, thereby preventing the neglect of compounds with potential anti-*M. tuberculosis* activity.

Notably, RTD displayed limited yet significant activity against nonreplicating *M. tuberculosis* (Figure 2). Additionally, partial synergistic effects were observed when RTD was combined with CLO, PTM, and TB47, resulting in enhanced inhibition of AlRa growth (Table 3). These findings collectively suggest that RTD may be a promising candidate for antimycobacterial drug development. We employed gavage administration and observed that RTD (the highest dosage of 25 mg/kg) exhibited no activity against *M. tuberculosis* and no obvious toxicity for mice (Figure 3A). The combination of RTD and TB47 did not demonstrate an obvious synergistic effect against *M. tuberculosis* in vivo via oral administration (Figure 3A). Therefore, we aimed to evaluate the in vivo efficacy of RTD using a recently developed murine model designed to enhance the pulmonary local concentration of drugs [26]. Our findings revealed that RTD, at a concentration of 1.56 mg/mL, displayed no in vivo anti-*M. tuberculosis* activity when administered via inhalation for ~25 min (Figure 3B). The efficacy of RTD in vivo is limited due to its potential inability to achieve a concentration sufficient for eliminating *M. tuberculosis*. This may be attributed to its imperfect chemical structure leading to suboptimal absorption and/or distribution. Despite its lack of anti-TB activity in vivo, RTD remains a valuable lead compound for the development of derivatives with improved in vitro and potential in vivo anti-*M. tuberculosis* properties.

Meanwhile, it proved challenging to obtain mycobacterial mutants resistant to RTD, highlighting its advantageous property in preventing the emergence of resistance. To address the challenge of identifying mutants resistant to RTD, we altered our strategy and conducted transcriptome analysis of AlRa treated with RTD to identify potential targets of RTD. We found that a total of 2329 genes had significantly altered expression levels (Figure 4A). Consequently, we established gene selection criteria based on a |log_2_Foldchange| greater than 2 and essential genes, resulting in the identification of eight genes (Table 4). Rv0351 along with Rv0352 act as cofactors that stimulate the ATPase activity of Rv0350 [33,34]. Rv0350 and Rv0440 play roles in maintaining *M. tuberculosis* protein stability and sustaining long-term cell vitality under stressful conditions [34]. Rv0384 is thought to be an ATPase subunit of an intracellular ATP-dependent protease crucial for *M. tuberculosis* survival within macrophages and maintaining dormant states [35]. Furthermore, Rv2720, involved in the SOS response, can bind to the operator region of multiple genes, inhibiting their expression during normal growth conditions [36,37]. However, Rv2720 may dissociate from these operators, triggering the activation of these genes to support DNA repair processes [36]. The specific function of Rv2827, categorized as a conserved hypothetical protein, remains unknown. Some studies have revealed that Rv2827 may be a DNA-binding protein responsible for antitoxin expression required for growth [38,39]. A previous report has shown that Rv3260 functions as a transcriptional regulator associated with cell division [40]. Notably, seven out of eight essential genes discussed herein exhibited upregulation following treatment of *M. tuberculosis* with RTD, with the exception of *Rv3260c*. Further comprehensive investigations are necessary to elucidate RTD’s mechanism of action, primarily targeting the aforementioned eight genes.

## 5. Conclusions

The present study encounters several limitations. Firstly, the potential targets of RTD or the genes associated with mycobacterial resistance to RTD remain unidentified. Only a handful of potential genes have been derived from transcriptome analysis, necessitating further verification and exploration in subsequent studies. Secondly, augmenting the in vivo antimycobacterial activity poses a critical and challenging issue that demands immediate attention. When administered via inhalation, the use of pro-solvents to enhance the solubility of RTD could potentially elevate its effective concentration in the lungs. Thirdly, although RTD has demonstrated antimycobacterial activity in vitro, this effect has not been observed in vivo. Therefore, RTD may serve as a promising lead compound for enhancing antibacterial activity both in vitro and in vivo through the design of optimized derivatives.

Despite certain limitations, our research has revealed that RTD displays inhibitory activities against specific mycobacterial strains, particularly *M. tuberculosis* and *M. marinum*, marking the first instance of such findings. RTD demonstrated a significant partial synergistic effect against *M. tuberculosis* in vitro when combined with CLO, PTM, and TB47, with FICIs ranging from 0.5 to 1. The screening for mycobacterial mutants resistant to RTD proved challenging during the experiment. This characteristic of RTD demonstrates favorable property as a clinical antimycobacterial agent that can prevent the emergence of drug-resistant strains. All results indicated that RTD provides fundamental information for drug repurposing or a lead compound for optimization and offers alternative options for clinical treatment. Further experiments are necessary to elucidate the antimycobacterial activity of RTD-related derivatives in vivo using various mice models and uncover the corresponding mechanism.

## 6. Patents

Based on our experimental results, we have applied for a Chinese invention patent with the patent number of 202310121229.X.

## Figures and Tables

**Figure 1 pharmaceutics-16-01049-f001:**
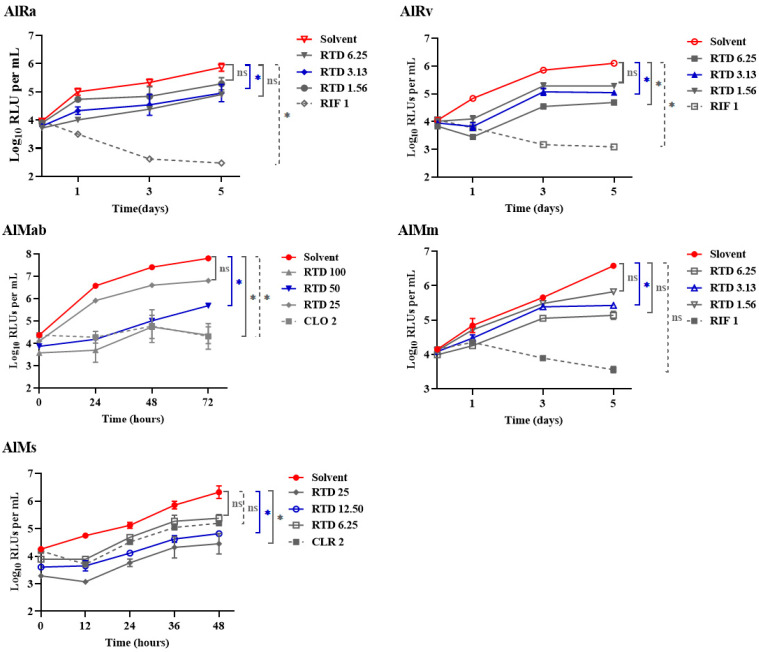
Time-killing curves of different mycobacteria treated with RTD. AlRa, autoluminescent *M. tuberculosis* H37Ra; AlRv, autoluminescent *M. tuberculosis* H37Rv; AlMm, autoluminescent *M. marinum*; AlMab, autoluminescent *M. abscessus*; AlMs, autoluminescent *M. smegmatis*; Solvent, dimethyl sulfoxide; RTD, rupatadine; RIF, rifampicin; CLO, clofazimine; CLR, clarithromycin; ns, *p* > 0.05; *, *p* < 0.05. The numbers following compounds indicate the corresponding concentrations (μg/mL).

**Figure 2 pharmaceutics-16-01049-f002:**
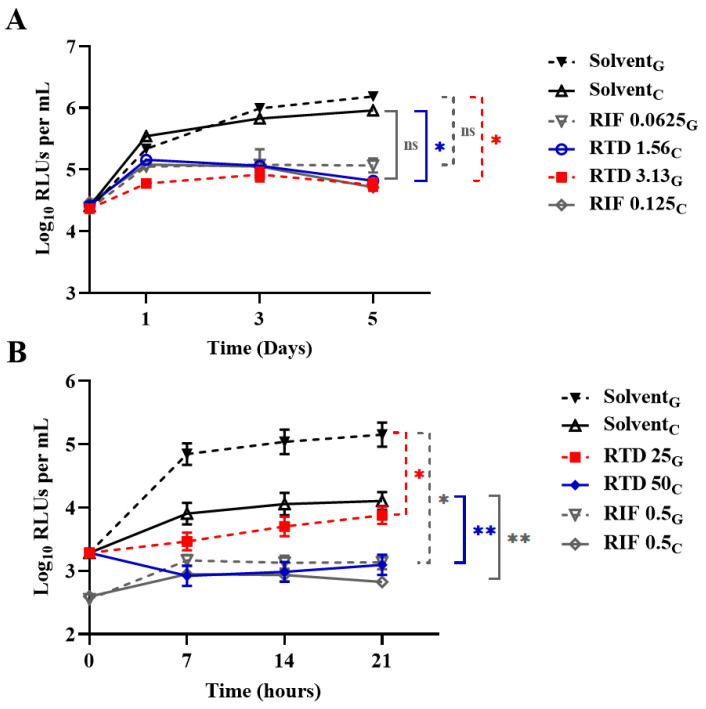
Time-killing curves of replicating AlRa or recovering curves of nonreplicating AlRa treated with RTD in diverse media. (**A**) Aerobic conditions; (**B**) anaerobic conditions. Solvent, dimethyl sulfoxide; RTD, rupatadine; RIF, rifampicin; ns, *p* > 0.05; *, *p* < 0.05; **, *p* < 0.01; G, 7H9 enriched with glycerol; C, 7H9 enriched with cholesterol. The numbers following RTD or RIF indicate the corresponding concentrations (μg/mL).

**Figure 3 pharmaceutics-16-01049-f003:**
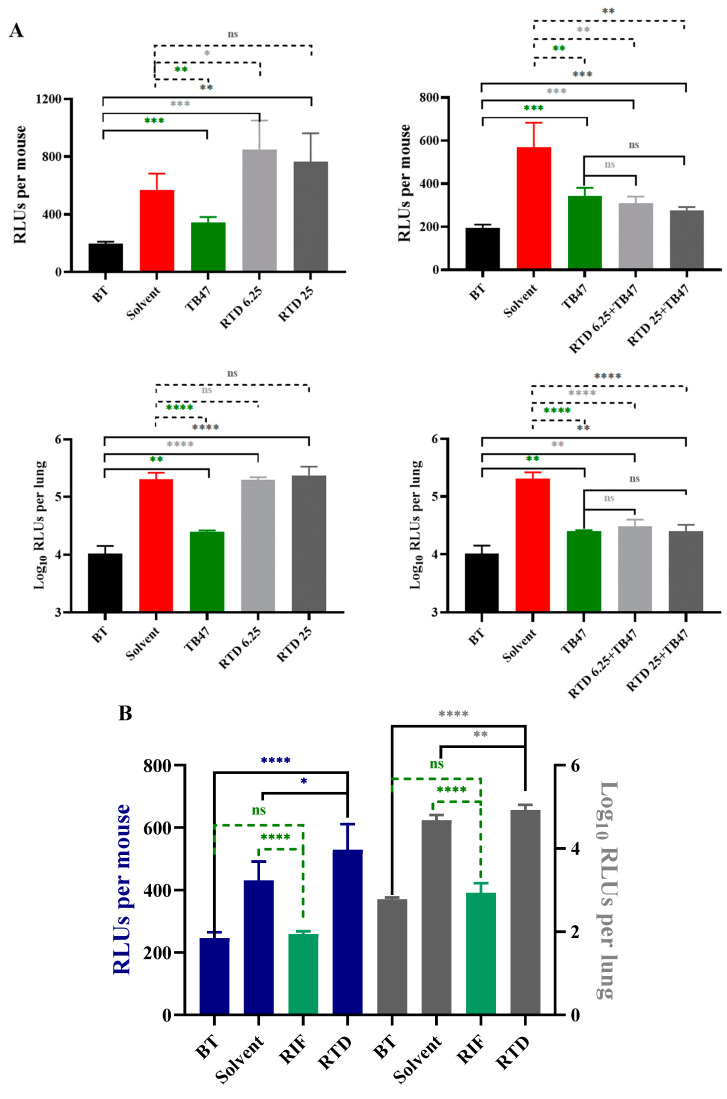
The in vivo activity of RTD against *M. tuberculosis*. (**A**) The in vivo activity of RTD and the combination of RTD and TB47 using oral administration. Mice were infected with AlRv via aerosol. The treatments were administered daily from the eighth day post-infection and lasted for eight days. The RLUs of live mice and the lung suspension were detected at initiation and one day after completion of treatment. The dosages of RTD were 6.25 and 25 mg/kg, and the dosage of TB47 was 25 mg/kg. (**B**) The in vivo activity of RTD using inhalable administration. Mice were infected with AlRv via aerosol. The treatments were administered daily from the next day post-infection and lasted for 15 days. The RLUs of live mice and the lung suspension were detected at initiation and one day after completion of treatment. The dosages of RTD and RIF were 1.56 and 2 mg/mL, respectively. The duration of inhalable administration (4 mL) each time was approximately 25 min. BT, before treatment; Solvent (A), sodium carboxymethyl cellulose; Solvent (B), distilled water; RTD, rupatadine; RIF, rifampicin; ns, *p* > 0.05; *, *p* < 0.05; **, *p* < 0.01; ***, *p* < 0.001; ****, *p* < 0.0001.

**Figure 4 pharmaceutics-16-01049-f004:**
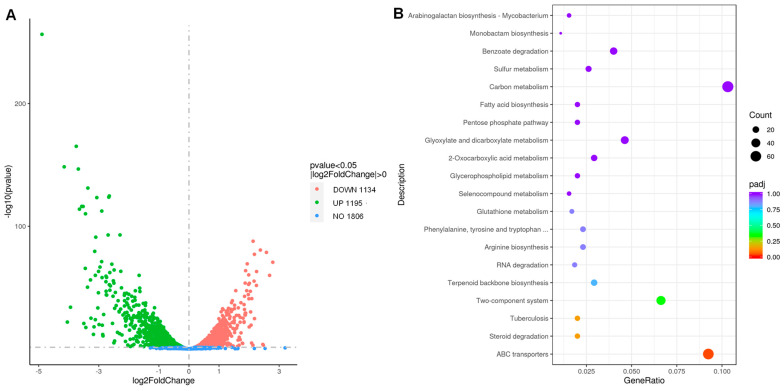
The transcriptome profile of *M. tuberculosis* cocultured with or without RTD treatment. (**A**) Volcano plot of genes with |log_2_FoldChange| > 0 and *p* < 0.05; (**B**) KEGG analysis of gene descriptions.

**Table 1 pharmaceutics-16-01049-t001:** The MICs of RTD against different mycobacteria.

Strains ^#^	MICs	
	μg/mL	μM
AlRa	3.13	7.52
AlRv	3.13	7.52
AlMm	3.13	7.52
AlMab	50	120.20
AlMs	12.5	30.05

^#^ AlRa, autoluminescent *M. tuberculosis* H37Ra; AlRv, autoluminescent *M. tuberculosis* H37Rv; AlMm, autoluminescent *M. marinum*; AlMab, autoluminescent *M. abscessus*; AlMs, autoluminescent *M. smegmatis*; RTD, rupatadine.

**Table 2 pharmaceutics-16-01049-t002:** The MICs of RTD against replicating and nonreplicating AlRa in diverse media.

Strains ^#^		MIC	
		μg/mL	μM
Replicating	AlRa ^G^	3.13	7.52
	AlRa ^C^	1.56	3.75
Nonreplicating	AlRa ^G^	25	60.10
	AlRa ^C^	50	120.20

^#^: AlRa, autoluminescent *M. tuberculosis* H37Ra; RTD, rupatadine; ^G^ and ^C^ following AlRa represent 7H9 enriched with glycerol and cholesterol, respectively.

**Table 3 pharmaceutics-16-01049-t003:** Activity of RTD in combination with anti-*M. tuberculosis* drugs against AlRa.

Drug *	MIC ^#^ (µg/mL)				FICI ^※^	Effects
	MIC _RTD_ ^C^	MIC _RTD_ ^A^	MIC ^C^	MIC ^A^		
AMK	0.20	3.13	0.5	0.5	1.0625	Indifferent
CLO	0.78	3.13	0.1	0.2	0.75	Partial synergistic
INH	3.13	3.13	0.003125	0.05	1.0625	Indifferent
LEV	0.20	3.13	0.125	0.125	1.0625	Indifferent
LZD	3.13	3.13	0.5	1	1.5	Indifferent
PTM	0.20	3.13	0.0625	0.125	0.5625	Partial synergistic
RIF	3.13	3.13	0.0015625	0.0125	1.125	Indifferent
STR	1.56	3.13	2	4	1	Additive
TB47	0.78	3.13	0.00075	0.0015	0.75	Partial synergistic

* AMK, amikacin; CLO, clofazimine; INH, isoniazid; LEV, levofloxacin; LZD, linezolid; PTM, pretomanid; RIF, rifampicin; STR, streptomycin; RTD, rupatadine. ^#^ MIC ^A^, the MIC of drug alone; MIC ^C^, the MIC of drug combined. **^※^** FICI, Fractional Inhibitory Concentration Index; FICI ≤ 0.5, synergistic; 0.5 < FICI < 1, partial synergistic; FICI = 1, additive; 1 < FICI ≤ 4, indifferent; FICI > 4, antagonistic [24].

**Table 4 pharmaceutics-16-01049-t004:** The summary of essential genes with |log_2_FoldChange| exceeding 2.

Gene_Id	log_2_FoldChange *	Gene_Name	Product/Function
*Rv0350*	3.68124885	*dnaK*	Probable chaperone protein DnaK (heat shock protein 70); acts as a chaperone.
*Rv0351*	3.750319008	*grpE*	Probable GrpE protein (HSP-70 cofactor); stimulates, jointly with Rv0352, the ATPase activity of Rv0350.
*Rv0352*	3.066383388	*dnaJ1*	Chaperone protein DnaJ (HSP-70 cofactor); stimulates, jointly with Rv0351, the ATPase activity of Rv0350.
*Rv0384c*	2.667760091	*clpB*	Probable endopeptidase ATP binding protein ClpB (heat shock protein F84.1); thought to be an ATPase subunit of an intracellular ATP-dependent protease.
*Rv0440*	2.448531361	*groEL2*	Molecular chaperone GroEL; prevents misfolding and promotes the refolding and proper assembly of unfolded polypeptides generated under stress conditions.
*Rv2720*	2.27206078	*lexA*	Repressor LexA; involved in regulation of nucleotide excision repair and sos response.
*Rv2827c*	2.142454829	*Rv2827c*	Hypothetical protein; unknown.
*Rv3260c*	-2.134883679	*whiB2*	Probable transcriptional regulatory protein WhiB-like WhiB2; involved in transcriptional mechanism.

* log_2_FoldChange > 0 indicates upregulation of the gene; log_2_FoldChange < 0 indicates downregulation of the gene.

## Data Availability

Data are contained within the article.

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
