# Peer review of "Assessment of the Efficacy of the Antihistamine Drug Rupatadine Used Alone or in Combination against Mycobacteria"

_pharmaceutics, 2024, doi:10.3390/pharmaceutics16081049_

Round 1

Reviewer 1 Report

Comments and Suggestions for Authors

The article reports a novel antimycobacterial agent chemotype identified through the drug repurposing of an antiallergic (antihistamine) compound. The results are solid, promising to open new avenues for combatting the global tuberculosis threat. The article can be published after a MINOR revision addressing mostly some minor presentation issues.

1) While the authors should certainly be congratulated for finding a promising novel chemotype, the story behind this discovery is obviously intriguing and will be very interesting for the readers. Was this compound identified through a screening campaign or some directed approaches?

2) To facilitate the efficacy evaluation and comparison between different drugs, it would be useful to list the molar (μM) drug concentrations in addition to the mass-based (μg/mL) values.

3) The results concerning the in vivo activity of RTD (or lack thereof) should be presented more carefully. Presently, the phrases like “RTD… displayed no in vivo anti-M. tuberculosis activity… These results imply that RTD may serve as a lead compound for the development of derivatives with improved in vivo anti-M. tuberculosis activity” look not quite logical. The possible reasons for the difference could perhaps be discussed.

4) English in the article is very good, but a little bit of editing would be useful in some places. E.g., “mice were administrated”, “inaugural inquiry”.

Comments on the Quality of English Language

English in the article is very good, but a little bit of editing would be useful in some places. E.g., “mice were administrated”, “inaugural inquiry”.

Author Response

Reviewer 1:

Comments and Suggestions for Authors

The article reports a novel antimycobacterial agent chemotype identified through the drug repurposing of an antiallergic (antihistamine) compound. The results are solid, promising to open new avenues for combatting the global tuberculosis threat. The article can be published after a MINOR revision addressing mostly some minor presentation issues.

Author’s response: We express our gratitude to you for the positive feedback on our manuscript.

1) While the authors should certainly be congratulated for finding a promising novel chemotype, the story behind this discovery is obviously intriguing and will be very interesting for the readers. Was this compound identified through a screening campaign or some directed approaches?

Author’s response: I would like to express my gratitude for your acknowledgement of our findings. Initially, we acquired an FDA-approved compound library comprising approximately 2000 clinical and pre-clinical drugs. Subsequently, we employed the autoluminescent M. tuberculosis H37Ra strain to primarily screen drugs with potential anti-tuberculosis activity, using only three concentrations (100, 10, and 1 μg/mL). Notably, we discovered that RTD exhibited remarkable anti-mycobacterial activities and further investigated its efficacy alone or in combination against mycobacteria. We have incorporated the relevant information into the revised submission of our manuscript.

2) To facilitate the efficacy evaluation and comparison between different drugs, it would be useful to list the molar (μM) drug concentrations in addition to the mass-based (μg/mL) values.

Author’s response: We value your insightful feedback. Following your recommendation, we have modified the sections pertaining to μM drug concentration in Tables 1 and 2 of our updated manuscript.

3) The results concerning the in vivo activity of RTD (or lack thereof) should be presented more carefully. Presently, the phrases like “RTD… displayed no in vivo anti-M. tuberculosis activity. These results imply that RTD may serve as a lead compound for the development of derivatives with improved in vivo anti-M. tuberculosis activity” look not quite logical. The possible reasons for the difference could perhaps be discussed.

Author’s response: Thank you for your insightful comments on our manuscript. Although RTD does not exhibit in vivo anti-tuberculosis activity, either through conventional oral administration (25 mg/kg) or inhalation administration (1.56 μg/mL), it still holds significant potential for serving as a reference in the discovery of anti-tuberculosis drugs. The in vivo efficacy of RTD is limited due to its potential inability to achieve a concentration sufficient for eliminating M. tuberculosis, which may be attributed to its imperfect chemical structure leading to suboptimal absorption and distribution. Despite its lack of anti-tuberculosis activity in vivo, RTD remains a valuable lead compound for the development of derivatives with improved in vitro and potential in vivo anti-M. tuberculosis properties. We updated the details marked in red in the revised manuscript.

4) English in the article is very good, but a little bit of editing would be useful in some places. E.g., “mice were administrated”, “inaugural inquiry”.

Author’s response: We sincerely appreciate your recognition of our English writing. We have meticulously reviewed and revised our manuscript, with particular attention to the sections you mentioned.

Reviewer 2 Report

Comments and Suggestions for Authors

1.       Authors need to include the statistical analysis in all the figures and figure legends along with the details of controls.

2.       References are missing in some of the places in materials and methods. Please update.

3.       How do authors determine the synergistic or additive activity of drugs with RTD against MTB? Please justify and discuss. Here is and reference to follow, https://pubmed.ncbi.nlm.nih.gov/27267959/.

4.       Authors should use a pathogenic strain of MTB. Please discuss and justify.

5.       There are many limitations associated with the study, please include a separate section describing the limitations of the current study.

Author Response

Reviewer 2:

Comments and Suggestions for Authors

1.Authors need to include the statistical analysis in all the figures and figure legends along with the details of controls.

Author’s response: We are grateful for your insightful suggestion for our manuscript. In light of the valuable feedback you provided, we have incorporated the statistical significance into the graphs. The details are highlighted in red in the updated manuscript.

  1. References are missing in some of the places in materials and methods. Please update.

Author’s response: We appreciate the reviewer's insightful comments on our manuscript. We have thoroughly revised and updated the references in the materials and methods section, providing a clearer depiction of this aspect.

3.How do authors determine the synergistic or additive activity of drugs with RTD against MTB? Please justify and discuss. Here is and reference to follow, https://pubmed.ncbi.nlm.nih.gov/27267959/.

Author’s response: As referenced in our manuscript (https://doi.org/10.1016/j.ejmech.2018.11.054), the synergistic and additive activities can be distinguished by comparing the FICI values. Different from the article you cited, our manuscript provides a more detailed categorization of the FICI values as follows: synergistic (FICI ≤ 0.5), partially synergistic (0.5 < FICI < 1), additive (FICI = 1), irrelevant (1 < FICI ≤ 4), and antagonistic (FICI > 4). We appreciate your advice for revising our manuscript, and we have incorporated the reference you mentioned above into our resubmitted manuscript in the “Materials and Methods” section to enhance the clarity of our study.

  1. Authors should use a pathogenic strain of MTB. Please discuss and justify.

Author’s response: We appreciate your valuable feedback on our manuscript. We have indeed determined the MICs of RTD against several pathogenic strains of M. tuberculosis isolated from Guangzhou Chest Hospital. However, RTD did not demonstrate significant anti-mycobacterial activity, with MICs as high as 50 μg/mL. These results are presented in the “3.1 Anti-mycobacterial Activity of RTD” section of the updated manuscript.

  1. There are many limitations associated with the study, please include a separate section describing the limitations of the current study.

Author’s response: We would like to express our gratitude for the valuable comments you have provided on our manuscript. The present study encounters several limitations. Firstly, the potential targets of RTD or the genes associated with mycobacterial resistance to RTD remain unidentified. Only a handful of potential genes have been derived from transcriptome analysis, necessitating further verification and exploration in subsequent studies. Secondly, augmenting the in vivo anti-mycobacterial activity poses a critical and challenging issue that demands immediate attention. When administered via inhalation, the use of pro-solvents to enhance the solubility of RTD could potentially elevate its effective concentration in the lungs. Thirdly, although RTD has demonstrated anti-mycobacterial activity in vitro, this effect has not been observed in vivo. Therefore, RTD may serve as a promising lead compound for enhancing antibacterial activity both in vitro and in vivo through the design of optimized derivatives. We have updated these limitations in the conclusion section of our revised manuscript.

Reviewer 3 Report

Comments and Suggestions for Authors

The presented study is novel, interesting and impressive.

Written with very good English, reads well.

I actually have no critical remarks.

My only remark is:

l. 25,71,130 etc. – it is unclear what TB47 is. Please explain.

Well done.

Author Response

Reviewer 3:

Comments and Suggestions for Authors

The presented study is novel, interesting and impressive. Written with very good English, reads well. I actually have no critical remarks. My only remark is: l. 25,71,130 etc. – it is unclear what TB47 is. Please explain. Well done.

Author’s response: I am grateful for your acknowledgement of our findings and the quality of our English writing. TB47 is an imidazopyridine amide antibiotic targeting the QcrB subunit within the cytochrome bc1 oxidase complex of electron transfer chain. By blocking this complex, TB47 reduces the intracellular ATP level, ultimately inhibiting the growth of M. tuberculosis. The explanation of TB47 is provided in the “2.2 Antimicrobials” section. Meanwhile, we have thoroughly revised our manuscript with particular attention to the sections you emphasized.
